Integrative omics analysis of plant-microbe synergies in petroleum pollution remediation

Mu Yi-Qian 1 2
Song Jian-Bo 3
Zhao Min 1 2
Ren Peng 1 2
Liu Han-Yu 1 2
Huang Xuan xuanhuang@nwu.edu.cn 3
1 National Engineering Laboratory for Exploration and Development of Low Permeability Oil and Gas Fields , Shaanxi , Xi’an , China
2 China National Petroleum Corporation Changqing Oilfield Branch Oil and Gas Technology Research Institute , Shaanxi , Xi’an , China
3 College of Life Sciences, Northwest University , Shaanxi , Xi’an , China
Mehmood Muhammad Aamer
Electronic publication date: 2025 Jun 20
Publication date: 2025
Volume: 13
Electronic Location ID: e19396
Received 2024 Nov 22; Accepted 2025 Apr 8
Copyright: ©2025 Mu et al.
Copyright year: 2025
Copyright holder: Mu et al.
License: This is an open access article distributed under the terms of the Creative Commons Attribution License, which permits unrestricted use, distribution, reproduction and adaptation in any medium and for any purpose provided that it is properly attributed. For attribution, the original author(s), title, publication source (PeerJ) and either DOI or URL of the article must be cited.
License URL: https://creativecommons.org/licenses/by/4.0/

Keywords: Petroleum hydrocarbon, Soil, Plant-microbial remediation, Rhizosphere microorganism, Omics analysis

Funding: The National Natural Science Foundation of China 31300223 Natural Science Foundation of Shaanxi Province 2016JM3001 Opening Foundation of National Engineering Laboratory and Development of Low Permeability Oil and Gas Fields KFKT2023-04 This study was supported by grants from the National Natural Science Foundation of China (31300223), Natural Science Foundation of Shaanxi Province (2016JM3001), Opening Foundation of National Engineering Laboratory and Development of Low Permeability Oil and Gas Fields (KFKT2023-04). The funders had no role in study design, data collection and analysis, decision to publish, or preparation of the manuscript.

==============================
As the petrochemical industry continues to advance, the exacerbation of ecological imbalance and environmental degradation due to petroleum pollution is increasingly pronounced. The synergistic interaction between plants and microorganisms are pivotal in the degradation of petroleum hydrocarbons; however, the underlying degradation mechanisms are not yet fully understood. This study aims to contribute to understanding these mechanisms by employing a multi-omics approach, integrating transcriptomics, 16S rRNA gene sequencing, and metabolomics, to analyze key differential genes, dominant microbial strains, and root-secreted metabolites involved in petroleum hydrocarbon degradation in alfalfa. Our findings revealed that several stress-related genes are upregulated in alfalfa contaminated with petroleum hydrocarbon. Moreover, Pseudomonas, Rhodococcus, and Brevundimonas were identified as dominant species in the rhizosphere microbiome. Metabolomics analysis identified pantothenic acid, malic acid, and ascorbic acid as critical metabolites that enhance hydrocarbon degradation. Application of pantothenic acid in oil-contaminated soil increased the degradation rate by approximately 10% compared to other treatments. These results highlight the potential of alfalfa-based phytoremediation strategies and offer a novel perspective for improving the efficiency of soil decontamination. Further research is needed to validate the scalability of these strategies for practical applications.

Introduction

Petroleum pollution has become a critical environmental and societal challenge, driven by industrial expansion and the urgent need for sustainable ecological solutions (Chen et al., 2019; Mojiri et al., 2019; Honda & Suzuki, 2020; Chau et al., 2023). Effective strategies for the degradation and remediation of hydrocarbon pollutants are essential to mitigate their adverse impact on ecosystems and human health (Robichaud et al., 2019; Suárez-Moo et al., 2020; Daâssi & Qabil Almaghribi, 2022). Among these strategies, phytoremediation leveraging plant-microbe interactions has emerged as a promising approach. This method capitalizes on the synergistic relationship between plants and microorganisms to enhance the breakdown of organic pollutants in contaminated soils and sediments (Chen et al., 2018; Włóka et al., 2019; Hou et al., 2021). Despite its potential, the specific rhizosphere microorganisms involved in the degradation of petroleum hydrocarbons (PHCs) and their interactions with plant metabolites, particularly in alfalfa, remain insufficiently understood. Addressing this gap requires multi-omics analyses to identify potential microbial candidates and effective enhancers that can optimize the remediation process, paving the way for their application in field-scale petroleum pollution management through well-designed optimization systems. Previous studies have demonstrated that the remediation of petroleum pollution through plant-microbe interactions is more effective compared to remediation approaches that utilize plants or microbes independently (Xu et al., 2014; Rehman et al., 2019). Certain root exudates from plants can enhance microbial proliferation and accelerate the degradation of petroleum hydrocarbons (Chaudhry et al., 2005; Khan et al., 2013; Rehman et al., 2019).

Effective plant-based remediation requires that plants exhibit a sufficient level of tolerance to specific pollutants, enabling them to effectively metabolize contaminants (Gkorezis et al., 2016; da Silva Correa et al., 2022; Abbas et al., 2023). Various plant species, such as Lolium perenne, Festuca arundinacea, Zea mays, Salix spp and Medicago sativa, have been used to remediate oil-contaminated soils (Wiltse et al., 1998; Chaîneau, Morel & Oudot, 2000; Huang et al., 2004; Yousaf et al., 2010; Yu et al., 2011; Yergeau et al., 2015). Meanwhile, over 100 genera of microorganisms known to degrade PHCs in soil have been identified, including fungi such as Mortierella, Aspergillus and Penicillium, and bacteria such as Nocardia, Pseudomonas, Rhodococcus (Rahman et al., 2003; Brooijmans, Pastink & Siezen, 2009; Glick, 2010). Xu et al. (2014) explored microbial remediation, phytoremediation, and their combination for treating contaminated soils. They found that the combined remediation system using Kocuria spp. and Lolium perenne achieved the most significant degradation of oil-contaminated soils (Xu et al., 2014). In this process, plants may contribute indirectly to remediation by creating favorable conditions for the growth of root-associated microorganisms.

Certain plant root systems secrete organic compounds that enhance microbial abundance and diversity in vegetated soils (Glick, 2010; Uroz et al., 2010; Pishchik et al., 2021). These microorganisms utilize the secreted compounds as sources of carbon and nitrogen (Chaudhry et al., 2005; Saravanan et al., 2020). Furthermore, plant roots release substances similar to PHCs, including terpenes, flavonoids, and various lignin-derived components, which may stimulate the expression of PHC-degrading genes in rhizospheric microorganisms (Sun et al., 2010). For example, phenolic compounds in root exudates have been shown to promote the growth of Mycobacterium, thereby accelerating the degradation of pyrene and benzo[a]pyrene (Toyama et al., 2011). Therefore, investigating root exudates can identify additional microorganisms capable of degrading PHCs.

In this study, alfalfa was initially transplanted into hydroponic solution containing various alkanes and performed transcriptomic analysis to identify a large number of DEGs. These genes may play crucial roles in enhancing plant tolerance to alkanes or facilitating their degradation. Subsequently, a 16S rRNA analysis of the alfalfa-cultivated soil identified Pseudomonas, Rhodococcus, and Brevundimonas as potentially dominant alkane-degrading strains. Certain rhizosphere microorganisms with alkane-degrading capabilities may thrive by utilizing compounds secreted by plant roots as sources of carbon and nitrogen. Consequently, a metabolomic analysis of the alfalfa hydroponic solution was conducted identifying metabolites that may support microbial survival. These findings establish a foundation for screening of plants and microorganisms involved in the degradation of PHCs.

Materials and methods

Plant materials and growth conditions

Alfalfa seeds were used as the experimental material. The seeds were soaked in dH2O overnight and then planted into soil (soil: vermiculite = 1:1). The cultivation conditions were as follows: ambient temperature 25 °C, relative humidity 60%, 16 h light (photon flux density 200 μmol/m2/s)/ 8 h dark cycle (Yang et al., 2022).

Simulated oil pollution

After growing for about 20 days, the alfalfa seedlings (Jin Huanghou) were transplanted into water cultivation containing a simulated saturated aliphatic hydrocarbon pollution mixed solution with a concentration of 1% for stress treatment (Dodecane: Cetane: Tetracosane=1:1:1). A half-strength Hoagland nutrient solution was added to support their growth (Lou et al., 2018). The alfalfa seedlings were collected at 0 h, 6 h, and 24 h after exposure to stress. The plants were ground into powder using liquid nitrogen and stored at −80 °C. Each treatment group was replicated three times.

RNA-sequencing analysis

Sequencing libraries and RNA sequencing were performed by the Meiji Biomedical Technology (Shanghai, China) using an Illumina Novaseq 6000 platform. To ensure the accuracy of subsequent bioinformatics analysis, fastp software was used to filter the raw sequencing data and obtain clean data. The quality-controlled data was aligned to the reference genome using HISAT2 software in order to obtain the mapped reads necessary for further analysis (Reference gene source: Medicago sativa; Reference genome version: Zhongmu No. 1) (Kim, Langmead & Salzberg, 2015). Mapped reads were assembled using Cufflinks software, and the results were compared with known transcripts for functional annotation (Trapnell et al., 2010).

Differential expression analysis of three groups (three biological replicates per condition) was performed using the DESeq2 package. The differentially expressed genes were identified with the criteria set as Padjust <0.05 and |log2FC| ≥ 1. Annotation analysis of the transcripts of DEGs was performed in the GO and KEGG databases (Klopfenstein et al., 2018).

16S rRNA gene sequence analysis

Total DNA was extracted from 5 g of each soil sample using E.Z.N.A.® soil DNA kit (Omega Bio-tek, Norcross, GA, US). The concentration and purity of DNA were assessed using a 1% agarose gel. Using the extracted DNA as a template, PCR amplification of the V3-V4 variable region of the 16S rRNA gene was performed with the upstream primer 338F: (5′-ACTCCTACGGGAGGCAGCAG-3′) and downstream primer 806R (5′-GGACTACHVGGG TWTCTAAT-3′) carrying barcode sequences (Liu et al., 2016). The PCR products from different samples were combined at an equal concentration and purified for sequencing on the Illumina PE300 platform. Chloroplast and mitochondrial sequences were removed from all samples using UPARSE v7.1 software (Edgar, 2013). The RDP classifier was used to compare the sequences with the Silva 16S rRNA gene database for taxonomic annotation of OTUs with a 50% confidence estimate (Wang et al., 2007). Additionally, functional prediction of 16S rRNA data was performed using PICRUSt2 (version 2.2.0) software (Douglas et al., 2020). A total of three groups of samples, with three biological replicates for each group. A: soil. B: A mixture of soil and oil. D-MX: A mixture of soil and oil used to grow alfalfa.

Metabolomics analysis

Metabolomics was performed as previously described with some modifications (Li et al., 2022). A total of 100 µL of concentrated hydroponic solution was transferred into a 1.5 mL centrifuge tube. Subsequently, 400 µL of extraction solution (acetonitrile: methanol = 1:1) containing 0.02 mg/mL of the internal standard L-2-chlorophenylalanine was added. After vortex mixing for 30 s, the sample was subjected to low-temperature ultrasonic extraction at 4 °C and 40 kHz for 30 min, followed by standing at −20 °C for an additional 30 min. The samples were then centrifuged at 4 °C, 13,000 g for 15 min. The supernatant was transferred and dried with nitrogen gas. The dried residues were dissolved in 100 µL solution (acetonitrile: water = 1:1) and then subjected to ultrasound extraction for 5 min at 4 °C and 40 kHz. After centrifugation at 4 °C and 13,000 g for 10 min, the supernatant was transferred to a sample vial for analysis by UHPLC -Q Exactive HF-X. The equal volumes of metabolites in all samples are mixed to prepare quality control (QC) samples, ensuring the reproducibility of the entire analysis process.

Samples were analyzed using electrospray ionization (ESI) in both positive and negative ion modes. The ESI source parameters were set as follows: Scan range (m/z), 70–1,050; Spray voltage (kV), 3.5 (ESI+) and 3.5 (ESI−); Heater temperature (∘C), 425; Capillary temperature (∘C), 325; Sheath gas flow rate (arb), 50; Auxiliary gas flow rate (arb), 13; Resolution (Full MS), 60,000; Resolution (MS/MS), 7,500; Normalized collision energy (eV), 20, 40, 60.

Data analysis

The raw data were processed using Progenesis QI v3.0 software (Waters Corporation, Milford, MA, USA), which performed baseline filtering, peak recognition, retention time correction, and peak alignment. This process generated a dataset containing retention time, mass-to-charge ratio (m/z), and peak intensity information. Metabolite identification was conducted by matching MS/MS spectral data in the HMDB database (http://www.hmdb.ca/) with an m/z tolerance of less than 10 ppm. The retention time and isotope similarity scores for all identified metabolites were provided in Table S7. Metabolic pathway enrichment analysis was performed using the KEGG database (http://www.genome.jp/kegg/), classifying the metabolites based on their involvement in specific pathways or biological functions.

Determination of total petroleum hydrocarbon degradation rate

A 4% mixture of petroleum hydrocarbons and soil was prepared, and the total petroleum hydrocarbon (TPH) content in the soil was assessed at various stages of alfalfa growth using a gravimetric method (Mishra et al., 2001). For each sample, 10 g of soil was subjected to sequential extraction with 100 mL each of hexane, dichloromethane, and chloroform. The combined extracts were evaporated to dryness under a nitrogen stream, and the residual TPH content was determined gravimetrically.

Statistical analyses

Data analysis and graph generation were performed using software program GraphPad Prism V. 7.0. One-way ANOVA was used for comparison of multiple data sets, with p < 0.05 as criterion for significant difference.

Results

The selection of the differentially expressed genes

The statistical power of this experimental design, calculated in RNASeqPower was 0.748 (three biological replicates). Principal component analysis (PCA) demonstrated distinct differences between the control and treatment groups by eliminating irrelevant data. The results indicated clear variances between these groups, and the data within each group exhibited good repeatability (Fig. 1A). Differentially expressed genes (DEGs) were selected using criteria of Padjust <0.05 and |log2FC| ≥ 1. After 6 h of simulated oil pollution stress, a total of 783 DEGs were identified, including 480 up-regulated and 303 down-regulated genes (Fig. 1B). After 24 h of stress, 582 DEGs were detected, with 434 up-regulated and 148 down-regulated genes (Fig. 1C). Venn diagram analysis revealed that 260 DEGs were common to both time points, with 523 unique DEGs after 6 h and 322 unique DEGs after 24 h (Fig. 1D). These results indicated that as the pollution time increased, the DEGs also differed.

Figure 1 The selection of the differentially expressed genes.

(A) Principal component analysis (PCA) scores plot of WR-6h, WR-24h and CK. (B), (C) Volcano plot of log2 fold change and –log10(P-value), for WR-6h vs. CK and WR-24h vs. CK. (D) Veen diagram of the number of shared and unique DEG in WR-6h vs. WR-24h, WR-6h vs. CK and WR-24h vs. CK. WR-24h: Alfalfa treated with simulated oil after 24 hours. WR-6h: Alfalfa treated with simulated oil after 6 h. CK: Alfalfa treated with H2O.

GO and KEGG enrichment analysis

Several DEGs were implicated in the resistance to or degradation of petroleum hydrocarbon pollutants. To elucidate the signaling pathways associated with these DEGs, GO analysis and KEGG analysis were performed. The GO analysis for the WR-6 h vs CK comparison revealed significant enrichment of DEGs in pathways related to the biosynthetic and metabolic process of salicylic acid, and the oxidation–reduction process (Fig. 2A). In the WR-24 h vs CK group, DEGs were significantly enriched in pathways involved in the synthesis of secondary metabolites, secondary metabolic processes, and glutathione metabolism (Fig. 2B).

Figure 2 GO and KEGG enrichment analysis.

(A), (B) KEGG enrichment analysis were shown in WR-6h vs. CK and WR-24h vs. CK. (C), (D) GO enrichment analysis were shown in WR-6 h vs. CK and WR-24h vs. CK.

The KEGG pathway analysis revealed distinct enrichment patterns in different time points. In the WR-6 h vs CK group, DEGs were predominantly enriched in the signal pathways of plant-pathogen interactions, glutathione metabolism, plant hormone signal transduction, and MAPKs signaling pathway (Fig. 2C). In the WR-24 h vs CK group, DEGs were mainly enriched in the signal pathways of oxidation–reduction process, carbon fixation in photosynthetic process and antenna proteins in photosynthesis (Fig. 2D). Therefore, these findings suggested that alfalfa may respond to petroleum hydrocarbon pollutants through various biosynthesis pathways, abiotic stress signaling transduction, and various oxidative-reductive metabolic pathways.

Quantitative analysis of DEGs

Transcriptome sequencing identified a total of 1,431 differentially expressed genes. Among these, nine most significantly DEGs in the antioxidant-related signaling pathway (peroxidase P7, glutathione S-transferase, transcription factor WRKY7), abiotic stress-related signaling pathway, and growth and development-related signaling pathway were selected for validation. The qRT-PCR results showed that the antioxidant-related genes (P7, ascorbate oxidase gene, glutathione S-transferase gene, REDOX2, P450, WRKY7) were all up-regulated, and the abiotic stress-related transcription factors (NAC and MYB1) also showed up-regulation (Fig. 3). At the same time, the growth development-related gene GA20ox1 was up-regulated, while the flower development-related gene FT was down-regulated (Fig. 3). The expression patterns of these nine genes are consistent with the sequencing results, thus proving the accuracy and repeatability of the transcriptome data.

Figure 3 Expression analysis of DEGs in different times.

One-way ANOVA was used to determine significant differences. *represented P < 0.05, **represented P < 0.01, ***represented P < 0.001. **** represented P < 0.0001.

Microbial diversity analysis of different soil samples

Transcriptomic analysis indicated that alfalfa exposure to petroleum pollutants activates its antioxidant enzyme system. Simultaneously, 16S rRNA sequencing was conducted to analyze the diversity and composition of the microbial community in the soil. Alpha diversity analysis employed various diversity indices to evaluate the abundance and richness of microbial communities in samples. PCA analysis indicated distinct variations between the control group (A: soil, B: a mixture of soil and oil) and the treatment group (D-MX: a mixture of soil and oil used to grow alfalfa), and the data within each group shows good repeatability (Fig. 4A). The Chao and Shannon indices revealed that the microbial abundance and richness in the A group were lower compared to those in the B and D-MX groups, with no significant differences observed between the B and D-MX groups (Figs. 4B, 4C). Venn diagram analysis identified 133 unique species present in the D-MX group (Fig. 4D). These special species may need rhizosphere exudates as carbon source or nitrogen source to maintain their survival.

Figure 4 Alpha and beta diversity analysis in A, B and D-MX.

(A) Principal component analysis (PCA) scores plot was shown to A, B and D-MX groups. (B), (C) Chao and Shannon indices were used to estimate the number of microbiotas in samples. (D) Veen diagram of the number of shared and unique ASV among the A, B and D-MX groups. A: soil. B: A mixture of soil and oil. D-MX: A mixture of soil and oil used to grow alfalfa.

Differential analysis of microbial community structure

Differential analysis of microbial community structure revealed the species present in each sample at various taxonomic levels and their relative abundance. Compared with the A and B group, family-level analysis showed that Pseudomonadaceae, Nocardiaceae, Rhizobiaceae, and Caulobacteraceae were significantly enriched in D-MX group (Fig. 5A). Furthermore, genus-level analysis showed that Pseudomonas and Rhodococcus were significantly enriched in D-MX group (Fig. 5B). The analysis of the community heatmap further indicated that Pseudomonas, Rhodococcus and Brevundimonas were the three most abundant microorganisms in the D-MX group (Fig. 5C). Besides, a comparison of the abundance of 15 species with the greatest differences revealed a significant decrease in the abundance of streptomyces, while the abundance of Pseudomonas, Rhodococcus and Brevundimonas showed a significant increase (Fig. 5D). These results suggested that the three dominant species in the D-MX group may have a role in the degradation of petroleum pollutants.

Figure 5 Differential analysis of the community structure of A, B and D-MX.

(A) The relative abundance of community structure at the family level for A, B and D-MX. (B) The relative abundance of community structure at the genus level for A, B and D-MX. (C) Comparison of the abundances of the bacteria of A, B and D-MX groups. (D) The heatmap analysis of the relative abundance in A, B and D-MX groups.

Metabolomic analysis of root exudates

Root exudates play a crucial role in supporting microbial ecosystems in the rhizosphere, and they are also essential for material cycling within this environment. To investigate whether alfalfa root exudates can degrade petroleum pollutants, a metabolomic analysis was conducted on the hydroponic solution where alfalfa was grown.

The KEGG pathway analysis results indicated that these metabolites were involved in amino acid metabolism, lipid metabolism, biosynthesis of secondary metabolites and carbohydrate metabolism (Fig. 6A). Additionally, the HMDB compound classification revealed that the metabolites were predominantly categorized as lipids and lipid-like molecules, organic acids and derivatives, organoheterocyclic compounds and organic oxygen compounds (Fig. 6B). Subsequently, alfalfa was transplanted into soils contaminated with various hydrocarbons, and the oil content was measured at different time. When pantothenic acid, malic acid, and ascorbic acid selected from metabolomics were applied to the contaminated soils where alfalfa was grown, significant improvements in degradation rates were observed. Notably, pantothenic acid application increased the degradation rate by approximately 10% compared to other treatments (Fig. 6C).

Figure 6 Metabolomic analysis of root exudates.

(A) The HMDB compound classification to root exudates. (B) The KEGG pathway analysis to root exudates. (C) The degradation rates after applying different substances to the oil sediment. T1: citric acid, T2: malic acid, T3: pantothenic acid.

Discussion

The soil-microbe-plant joint system plays an important role in the remediation of environmental pollutants, with its effectiveness relying on the soil environment, high-quality microbial resources, and resilient plants with robust root regulatory systems (Rezek et al., 2008; Peixoto, Vermelho & Rosado, 2011; Liu et al., 2022). Alfalfa, a perennial herbaceous plant in the legume family, exhibits strong cold resistance, salt and alkali tolerance, and adaptability to challenging environments (Ao et al., 2022; Wang et al., 2023; Zhao et al., 2023). Its extensive root system supports a diverse array of nitrogen-fixing and root-associated microorganisms, making it suitable candidate for the bioremediation process of various pollutants (Ye et al., 2014). Wiltse et al. (1998) used alfalfa for oil pollution remediation, achieving a degradation rate of around 46%. Further research reported that improved alfalfa varieties achieved petroleum degradation rates ranging from 33% to 56%. Moreover, studies have demonstrated that alfalfa alone can achieve a 35.21% degradation of TPHs, while combining alfalfa with compost improves plant growth and enhances TPH phytoremediation (Yousaf et al., 2022). In this study, alfalfa alone achieved a TPH degradation rate of 38%, while the addition of pantothenic acid significantly enhanced the rate to 67% (Fig. 6). Composting or applying specific compounds likely supports rhizosphere microorganism activity, thereby improving degradation efficiency. These findings highlight alfalfa’s potential as a key player in bioremediation efforts, offering a sustainable approach to addressing environmental contamination while enhancing soil health and ecosystem resilience. Future studies should focus on optimizing its interactions with specific microbial communities to further boost degradation efficiency.

Phytoremediation is a site-specific strategy for environmental remediation (Joner et al., 2004; Rojas-Solis, Larsen & Lindig-Cisneros, 2023). Understanding the molecular impacts of PHC on various plant species can provide more precise physiological data to guide plant selection (Afzal et al., 2019). Transcriptomic analysis of Arabidopsis thaliana exposed to PHCs revealed gene expression patterns resembling abiotic stress responses, including differential expression of genes related to heat, hypoxic, oxidative, and osmotic stress (Muniz Nardeli et al., 2016). Similarly, Amorpha fruticosa seedlings exposed to PHC-contaminated soil resulted in increased activities of glutathione reductase (GR), superoxide dismutase (SOD), and catalase (CAT), mitigating reactive oxygen species (ROS) accumulation (Cui et al., 2016). Assessment of these stress-responsive genes at the molecular level aids in the selection of more suitable plants for PHCs bioremediation. In this study, transcriptomic analysis of alfalfa revealed the upregulation of certain abiotic stress-related genes at both 6 h and 24 h post-exposure to stress (Fig. 1C). In the WR-6 h vs CK group, GO enrichment analysis showed early activation of pathways related to salicylic acid signaling, oxidative stress, and glutathione metabolism, indicating an immediate molecular response. In the WR-24 h vs CK group, secondary metabolite biosynthesis processes were predominantly enriched, suggesting a transition from signaling to metabolite synthesis to mitigate stress or degrade PHCs. Notably, DEGs were enriched in lignin biosynthetic and flavonoid biosynthesis pathways. Plants release organic compounds such as terpenes, flavonoids and some lignin-derived components, which share chemical similarities with PHCs and may induce the expression of PHC-degrading genes in rhizospheric microorganisms (Sun et al., 2010). The identification of these genes could inspire further improvements in alfalfa, facilitating the cultivation of new varieties with enhanced resistance and stronger ability to degrade petroleum pollutants.

Rhizoremediation, which combines phytoremediation and bioremediation, is an effective strategy for addressing complex pollution environments caused by petroleum and its derivatives (Lacalle et al., 2018). Previous studies have identified over 200 types of microorganisms capable of degrading PHCs, with the Rhodococcus, Pseudomonas, and Acinetobacter demonstrating particularly effective degradation (Huan et al., 2014). To investigate the microbial diversity in alfalfa root systems, we conducted 16S rRNA gene sequence analysis. Consistent with earlier findings, the abundance of Rhodococcus and Pseudomonas in the D-MX group is higher compared to A and B groups (Fig. 5A). These genera are known for their strong degradation capacities, which can be attributed to their possession of key alkane monooxygenase, including typical P450 (degrading C5-10 alkanes), AlkB (degrading C5-C16 alkanes), and AlmA (degrading C10-C30 alkanes). The enrichment of these microbes in the rhizosphere suggests that alfalfa root exudates may facilitate their survival and activity, further enhancing PHC degradation.

Figure 7 Integrated analysis of alfalfa-mediated PHC degradation mechanisms.

The study combines transcriptomics, 16S rRNA gene sequencing, and metabolomics to investigate alfalfa’s response to petroleum stress. Transcriptomics identifies DEGs associated with stress and degradation pathways, while 16S rRNA sequencing reveals dominant microbial strains in the rhizosphere. Metabolomics highlights key metabolites that potentially support microbial growth and enhance degradation. These findings contribute to understanding the synergistic mechanisms underlying PHC degradation in oil-contaminated soils.

Rhizoremediation is a promising method for the removal of PHCs from petroleum-contaminated soils, involving the collaboration among plants, microorganisms, and the rhizosphere environment. Integrating multi-omics analyses could provide deeper insights into the mechanisms underlying rhizosphere microbial community responses and PHC degradation processes (Zhang, Chen & Gong, 2024). To optimize remediation strategies, it is advisable to assess the environmental conditions and biotic factors at the petroleum-contaminated sites to better identify microbial communities capable of effectively degrading PHCs. However, limitations in techniques for cultivating complete microbial consortia, the experimental results regarding petroleum pollutant degradation have not yet been widely applied. Therefore, further investigation is warranted on how to translate findings from simulated petroleum pollution into practical applications in actual contaminated sites.

Conclusions

Based on the findings of this study, we conclude that the synergistic relationship between alfalfa and its rhizosphere microbiome plays a critical role in the degradation of PHCs, with specific plant metabolites acting as enhancers of microbial degradation pathways (Fig. 7). The observed upregulation of stress-related genes in alfalfa underscores its adaptive response to petroleum contamination, which may further support microbial activity and resilience. The identification of dominant microbial genera, including Pseudomonas, Rhodococcus, and Brevundimonas, along with the targeted application of metabolites like pantothenic acid, highlights promising pathways to accelerate PHC degradation. Notably, the application of pantothenic acid improved degradation rates by approximately 10%, demonstrating the potential of metabolic amendments to enhance phytoremediation efficiency.

While this study provides valuable insights into the mechanisms underlying plant-microbe interactions and the role of metabolites in phytoremediation, several limitations should be acknowledged. The study was conducted under controlled experimental conditions, which may not fully reflect the complexities of natural field environments. Future research should focus on scaling this approach for field applications, optimizing plant-microbe consortia under diverse environmental conditions, and assessing the ecological sustainability of these interventions.

Supplemental Information

Supplemental Information 1 MIQE checklist for qPCR methods and analysis

The first column indicates Item to check according to the MIQE guidelines. The second column indicates importance. E is short for essential while D is short for Desirable. The third column indicates Checklist in this study.

Supplemental Information 2 RNA Quality assessment

The first row (from B1 to G1) indicates Concentration, Volume, Yield, RIN (RNA Integrity Number), 28S/18S, and Quality Class of RNA, respectively. The first column (from A2 to A12) indicates 4 group samples (each contains 3 independent biological repeats, resulting in a total of 12 samples) in this study.

Supplemental Information 3 List of primers used in qPCR analysis

The first column (from A2 to A9) indicates Gene ID of 10 selected genes (including the reference gene) for qPCR validation. The first row (from B1 to G1) indicates Gene Name, Forward primer, Reverse Primer, , Tm, Product Size, Primer Blast result in NCBI, respectively

Supplemental Information 4 The raw data of the Cq values for qPCR

A2-A126 were collected samples A2-A126 were collected at different time points. The Cq values for the target gene and GAPDH are listed in columns B and C, respectively. Column D represents △Cq, column E represents the average, column F represents △△Cq, and column G represents 2-△△Cq.

Supplemental Information 5 Determination of Total Petroleum Hydrocarbon (TPH) Degradation Rate

The column A represents different times of samples, and columns B to M represent the TPH degradation Rate.

Supplemental Information 6 Metabolites identified in the positive mode

Supplemental Information 7 Metabolites identified in the negative mode

Supplemental Information 8 The retention time and isotope similarity scores for all identified metabolites

Additional Information and Declarations

Competing Interests

Author Contributions

Microarray Data Deposition

Data Availability

The authors declare there are no competing interests.

Yi-Qian Mu conceived and designed the experiments, prepared figures and/or tables, authored or reviewed drafts of the article, and approved the final draft.

Jian-Bo Song conceived and designed the experiments, prepared figures and/or tables, authored or reviewed drafts of the article, and approved the final draft.

Min Zhao performed the experiments, prepared figures and/or tables, and approved the final draft.

Peng Ren performed the experiments, authored or reviewed drafts of the article, and approved the final draft.

Han-Yu Liu analyzed the data, prepared figures and/or tables, authored or reviewed drafts of the article, and approved the final draft.

Xuan Huang analyzed the data, prepared figures and/or tables, authored or reviewed drafts of the article, and approved the final draft.

The following information was supplied regarding the deposition of microarray data:

We will later upload multiple omics data.

The following information was supplied regarding data availability:

The sequences are available at NCBI: PRJNA1187007, PRJNA1187032.

The metabolomic raw data is available at MetaboLights: MTBLS11700.

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
