# Peer review of "Integrative omics analysis of plant-microbe synergies in petroleum pollution remediation"

_PeerJ, doi:10.7717/peerj.19396_

## Round 0.1 · original submission · Major Revisions

Dear Authors

Please revise your manuscript carefully while addressing questions/comments provided by the reviewers.

We look forward to your revised submission in due course!

Reviewer 1 ·

Basic reporting

Overall, the paper was written in an easy-to-understand manner. However, there are still some sections that need to be improved. Hence, this paper must be revised carefully before being considered for publication. I hope the comments below will help improve the paper further.
Throughout the manuscript, there are a lot of grammatical and syntax errors that need to be corrected e.g. L127-128, L197-198, L265-266
Rewrite heading 3.4

Experimental design

- Please provide an additional figure to summarize the whole workflow.
What criteria were used to identify differentially expressed genes (DEGs) in the transcriptomic analysis?
How was the dominance of Pseudomonas, Rhodococcus, and Brevundimonas in the soil microbiome determined? Were appropriate controls included?
Were the metabolites identified through metabolomics cross-validated with microbial growth experiments to confirm their role in supporting microbial survival?
Was the alfalfa strain selected for this study representative of natural populations, or was it a specially bred variety?

Validity of the findings

The discussion section lacks a comprehensive literature review and does not adequately relate the findings to the existing body of literature.
How robust are the metabolomic findings? Could these metabolites be byproducts of alfalfa metabolism unrelated to alkane degradation?
Does the study provide sufficient evidence to conclude that the plant-microbe interaction is driving alkane degradation?
How scalable is this approach for field applications in contaminated soils or water?
What are the potential limitations or challenges in leveraging alfalfa and associated microbes for hydrocarbon remediation?
Could this study be extended to other plant species, and if so, which ones might be most promising?
What are the ecological risks of introducing or promoting specific microbial strains in natural environments?
How does this study contribute to the understanding of plant-microbe interactions in the context of environmental sustainability?

Additional comments

Abstract:
- Major revisions must be made before the main content is amended.
- An abstract is often presented separately from the article, so it must be able to stand alone. Hence, the study's problem statement, aim, novelty and results should all be included in one paragraph of the abstract.
Which knowledge gaps did you find? Highlight them in the concluding sentences.

Introduction:
The introduction needs a clearer and more concise first paragraph, effectively addressing the research gap. The problem statement should be strengthened, with a more detailed discussion to establish the context. Additionally, the societal and industrial relevance of the study must be emphasized

Conclusions
- Please include what was done in the study and the optimized results.
- Please include the limitations and what can be done in the future.

References
- Kindly revise the reference format according to the author's guidelines.
- Authors are encouraged to cite more recent and relevant literature from the target journal.

Reviewer 2 ·

Basic reporting

Overall, the manuscript is well-written and is of interest as it discusses an important aspect for aquatic bioremediation. The manuscript will be suitable for publication after "Major Revisions".

• Language of manuscript needs careful checking and revision. Pay special attention to the appropriate use of prepositions.
• Improve the resolution of figure 2 for better clarity and readability.
• Preferably references should be from last 3-5 years, except for the one’s referring to a method.
• Clearly state the research gap and novelty of the present study in the introduction.
• Avoid the use of personal pronouns like “he, she, we” etc.

Experimental design

• Revise the “Materials and methods” as it should be in past tense.
• Provide details of simulated pollution mixed solution.
• How much Hoagland nutrient solution was used?
• Correct the sentence structure of line 123-124 “a total of 100 uL of concentrated …… chlorophenylalanine”.

Validity of the findings

• At line 204, does only Pseudomonadaceae was significantly enriched or was there any other family? Check and correct the sentence accordingly.
• Revision of discussion section is recommended; it should be more elaborated and in-line with the findings of the present study.

---

## Round 0.2 · Minor Revisions

Although authors have improved their manuscript and reviewers have recommended this manuscript to be accepted, our Section Editor has asked for minor changes to further improve the manuscript and meet the journal's standards.

Editor's comments:
Please add the methodology and the criteria for the metabolomics part.
1. Which software and databases were used?
2. Which matching parameters (m/z tolerance, isotopic pattern fit, retention times)? 3. How were the metabolic pathways inferred (e.g. using MetaboAnalyst/ MummiChog/GSEA)?

Reviewer 1 ·

Basic reporting

OK

Experimental design

OK

Validity of the findings

OK

Reviewer 2 ·

Basic reporting

The authors incorporated all the suggestions. Manuscript meets the requirement of Journal and can be Accepted for publication.

Experimental design

-

Validity of the findings

-

Additional comments

-

---

## Round 0.3 · accepted · Accept

Thanks for submitting the updated version!